# Magic Numbers in Boson ^4^He Clusters: The Auger Evaporation Mechanism

**DOI:** 10.3390/molecules26206244

**Published:** 2021-10-15

**Authors:** Elena Spreafico, Giorgio Benedek, Oleg Kornilov, Jan Peter Toennies

**Affiliations:** 1Department of Materials Science, University of Milano-Bicocca, Via R. Cozzi 53, 20125 Milan, Italy; e.spreafico1@gmail.com; 2Donostia International Physics Center (DIPC), Paseo M. de Lardizàbal 4, 20018 Donostia-San Sebastián, Spain; 3Max-Born-Institute for Nonlinear Optics and Short Pulse Spectroscopy, Max-Born-Straße 2A, 12489 Berlin, Germany; oleg.kornilov@mbi-berlin.de; 4Max-Planck-Institute for Dynamics and Self-Organization, Bunsenstrasse 10, 37073 Göttingen, Germany; jtoenni@gwdg.de

**Keywords:** Van der Waals bonds, helium clusters, Auger evaporation, magic numbers, cluster supersonic jets

## Abstract

The absence of magic numbers in bosonic ^4^He clusters predicted by all theories since 1984 has been challenged by high-resolution matter-wave diffraction experiments. The observed magic numbers were explained in terms of enhanced growth rates of specific cluster sizes for which an additional excitation level calculated by diffusion Monte Carlo is stabilized. The present theoretical study provides an alternative explanation based on a simple independent particle model of the He clusters. Collisions between cluster atoms in excited states within the cluster lead to selective evaporation via an Auger process. The calculated magic numbers as well as the shape of the number distributions are in quite reasonable agreement with the experiments.

## 1. Introduction

Van der Waals interatomic bonds are the weakest in nature and, unlike covalent bonds, are isotropic, Thus the atomic clusters of heavy rare gases tend to grow in close-packed icosahedral shapes, with magic numbers determined by pure geometrical constraints [1]. Not so for ^4^He clusters: in view of their liquid and generally superfluid state, magic numbers would seem to be unlikely. Although some evidence of magic numbers in ^4^He clusters not corresponding to special geometries has been reported as early as 1983 [2], theoretical studies of cluster energies have indeed failed to provide any evidence for magic numbers. Several calculations based on different Monte Carlo methods [3,4,5,6,7] that the binding energy per atom ε(N) of small ^4^He clusters, and in general for boson clusters [8], is a smooth monotonic function of the atom number *N*. Nevertheless, high-resolution matter-wave diffraction experiments by Brühl et al. [7,9] have shown that the abundance *ρ*(*N*) of ^4^He clusters formed during a free jet supersonic expansion into vacuum from a cryogenic source is not monotonously decreasing for increasing *N*, as expected from the behavior of ε(N), but shows a series of maxima at certain *magic* numbers. The numbers *N* = 10, 14, 26, 44, ... at which maxima occur [9] do not correspond to any of the geometric magic numbers for the heavier rare gas clusters nor of the possible magic numbers of the ions formed in the detection process. It is however interesting that, in case of stronger polarization forces exerted by an ion on the surrounding He atoms, close-packed arrangements and therefore geometrical magic numbers are favored. Note that ionization occurs after the free jet clusters have reached thermodynamic equilibrium and cannot alter the number distribution. see, e.g., [10]. Rather they could be explained on the basis of a thermodynamic equilibrium model for the growth of the clusters [7]. Each time a new collective state with quantum numbers (*n*,*l*), obtained from a dedicated diffusion Monte Carlo calculation, is bound, a sudden jump in the ^4^He cluster partition function *Z*(*N*) occurs. As a consequence the equilibrium constant *K*(*N*) for the aggregation process ^4^He*_N−1_* + ^4^He ↔ ^4^He*_N_*, which is given by the function *Z*(*N*)*/Z*(*N* − 1), exhibits sharp peaks at *N* = *N_nl_* in good agreement with the experimental magic numbers. Thus, the enhanced growth of the magic number clusters reconciles the apparent disagreement with the predicted monotonous behavior of the binding energy per atom.

## 2. Theory

The present theory provides an alternative explanation by accounting for the role of the elementary processes of evaporation occurring in a supersonic cluster beam with a very high speed-ratio as in the matter wave experiments [9]. It is assumed that the final asymptotic cluster size distribution is a result of two sequential processes. First the clusters grow rapidly and, as a result of the energy of recombination, are internally hot; then, in a next step they cool down to their final temperature by evaporation of atoms [11,12,13].

Note that all these theories are based on thermodynamics and predict only a gradual dependence of the rate of evaporation on the cluster number size.

Since the densities and the relative velocities of the clusters and single atoms and consequently their collision probabilities in the final stages of the expansion are comparatively small, the kinetics are dominated by unimolecular evaporation of single atoms. As a result, the hot large clusters are transformed into smaller and cooler clusters until a final distribution *ρ*(*N*) stabilizes at some average temperature (0.37 K for ^4^He droplets [14]). The rate equation for the densities *ρ*(*N*) can be written as
(1)$$\begin{array}{c} \frac{d\rho (N)}{dt}=-P(N)\;\rho (N)+P(N+1)\;\rho (N+1)-\;\sum_{N\prime }\sigma (N,N\prime )\;\rho (N)\;\rho (N\prime )\\ +\sum_{N\prime <N}\sigma (N-N\prime ,N\prime )\;\rho (N-N\prime )\;\rho (N\prime )\; \end{array}$$
where *P*(*N*) is the rate of evaporation of a cluster consisting of *N* atoms and σ(N,N′) is the probability of coalescence of two clusters of size *N* and *N′*. The fission of large clusters to smaller ones is neglected in Equation (1). The asymptotic steady state conditions far downstream from the nozzle are obtained by setting dρ(N)/dt=0. Moreover, for clusters moving all at about the same speed the relative velocities corresponding to beam temperatures in the mK range [15] are so low that coalescence processes may be neglected. Under these conditions *ρ*(*N*) is, in a first approximation, inversely proportional to the ratio of the evaporation *P*(*N*).

Next it is assumed that an Auger process, involving two-body collisions, governs the evaporation of single atoms. As argued in the discussion below, phonon-induced evaporation is under the present physical conditions much less probable and is not considered. In the Auger process one bound He atom gains enough energy to leave the cluster in a collision with another atom, which drops from an initial excited state to a lower energy state (Figure 1). In order to investigate the basic physics of this process in boson clusters, a simplified independent particle model is considered in which a small number (*N <* 125) of ^4^He atoms is trapped in a spherical box potential of radius *R* = *R*(*N*) and depth $-V_{0}≅-\varepsilon (N)$. The model shows that the Auger evaporation rate strongly depends on the presence of bound one-particle states near the vacuum threshold, namely around those atom numbers at which a bunch of new bound states is created which are depleted compared to the adjacent cluster sizes. The role of bound states in determining the size distributions is similar to the analysis by Guardiola et al. [7] and suggests that Auger evaporation is actually the microscopic mechanism leading to the magic numbers.

In order calculate the rate of Auger collisions the wave functions of the atoms inside the clusters are required and for this reason an independent particle description is invoked. Although most current models of helium clusters are based on the diffusion Monte Carlo method (*T* = 0) and Path Integral Monte Carlo methods (*T* ≠ 0), or related algorithms, these models suffer from providing little information on the behavior of the individual constituents. The independent particle method, first introduced to ^4^He clusters and implemented via a variational Monte Carlo method by Schmid et al. in 1965 [16] and further refined by Lewart et al. [17] and Ramakrishna and Whaley [18], has been revived in connection with theories of Bose-Einstein condensed ultra-cold gases [19]. For calculating the collision dynamics between two bosons an exchange-symmetric two-atom wavefunction is required
(2)$$\psi_{ij}(r_{1},r_{2})=\frac{1}{\sqrt{2(1+\delta_{ij})}}[\phi_{i}(r_{1})\phi_{j}(r_{2})+\phi_{j}(r_{1})\phi_{i}(r_{2})]$$
where $\delta_{ij}$ is the Kronecker symbol. Since the single particles move freely inside the cluster they are approximated by a plane-wave representation, $\varphi_{j}(r)\propto \exp(ik_{j}\cdot r)$. Then the two-atom wavefunction can be expressed as a function of the internal coordinate $r=r_{2}-r_{1}$ and center-of-mass coordinate $r_{CM}=(r_{2}+r_{1})/2$:(3)$$\begin{array}{ll} \psi_{ij}(r_{1},r_{2})= & \frac{1}{N}\exp(i\;k_{CM}\cdot r_{CM})\;\cos(k\cdot r)\\ & =\frac{\sqrt{4\pi }}{N}\exp(ik_{CM}\cdot r_{CM})\sum_{n=0}^{\infty }{\left(-1\right)}^{n}{\left(4n+1\right)}^{1/2}j_{2n}(kr)\;Y_{2n,0}(\theta ) \end{array}$$
where $k=k_{j}-k_{i}$, $k_{CM}=(k_{j}+k_{i})/2$, *m** is the atomic effective mass, N a normalization constant, *θ* is the angle of **r** with respect to the conventional *z* axis. The atom-pair angular quantum number L≡2n, labelling the *m* = 0 spherical harmonics $Y_{L,0}(\theta )$ and the spherical Bessel functions $j_{L}(kr)$ in Equation (3), is an even integer originating from the composition of the individual atom angular momenta **l**_1_ and **l**_2_. If the eigenfunctions of Equation (2) are expressed via free-particle spherical waves, $\varphi_{klm}(r)\propto j_{l}(kr)Y_{l\;m}(\theta ,\;\varphi )$ and exchange is neglected, the matrix element $\langle \psi_{ij}|k_{j}\cdot k_{i}|\psi_{ij}\rangle =0$, which ensures that the expectation value of the atom-pair internal kinetic energy $\hbar^{2}\;k^{2}/4m^{*}$ is just $(E_{i}+E_{j})/2$.

The coupling leading to Auger processes is essentially given by the He-He interatomic potential, which is here chosen in the Tang-Toennies (TT) form [20,21] restricted to the first dipole-dipole dispersion term in the London expansion,
(4)$$V(r)=\left[A-\frac{C_{6}}{r_{}^{6}}\sum_{k=7}^{\infty }\frac{{\left(\beta \;r\right)}^{k}}{k!}\right]\;e^{-\beta \;r}$$
with $r=|\;r_{2}-r_{1}|$ the distance between atoms at positions **r**_1_ and **r**_2_, *A =* 22.16 a.u., *β* = 2.388 a.u., and *C*_6_ = 1.461 a.u. [22]. The transition rate encompassing the two processes of Figure 1 is expressed as
(5)$$\;W(E_{i},E_{j}\to E_{i\prime },E_{j\prime })=\frac{2\pi }{\hbar }{\left|\;\langle \psi_{i\prime j\prime }(r_{1},r_{2})|V(r)\;|\psi_{ij}(r_{1},r_{2})\rangle \;\right|}^{2}g(E_{i}+E_{j}-E_{j\prime })$$
where *g*(*E*) is the free-atom density of states. Due to the short range of V(r) as compared to the cluster diameter, surface effects are neglected.

Since the total energy is conserved ($E_{i\prime }+E_{j\prime }=E_{i}+E_{j}$) and also *L* is conserved due to the spherical symmetry of V(r), a diagonal matrix element occurs in Equation (5). Although the TT potential V(r) has no divergence, the Born approximation adopted in the Fermi golden rule of Equation (5) may not be sufficient due to the size of the repulsive potential, and distorted waves decaying exponentially for *r* smaller than the classical turning-point distance *d*_0_ should be used. This complication is avoided by restricting the integrations in Equation (5) to the range *d*_0_ ≤ *r* ≤ *R*_0_, with the minimum distance $d_{0}$ corresponding to the atomic co-volume $b=4\pi \;d_{0}^{3}/3$ = 40 Å^3^ (*excluded volume* approximation) and *R*_0_~*R*(*N*) the cluster radius, and using wavefunctions constructed on the basis of free-particle spherical waves.

In the present approximation the confinement due to the finite cluster radius effects acts through the discretization of energy levels $E_{j}\equiv E(n_{j},l_{j})$ appropriate to the cluster size, with *n_j_* and *l_j_* the radial and angular quantum numbers of the *j*-th level, respectively. An acceptable approximation for the energy eigenvalues of a spherical box of radius *R* and potential bottom −*V*_0_ is [23]
(6)$$E(n,l)≅-V_{0}+\frac{\pi^{2}\hbar^{2}}{8m^{*}R^{2}}{\left(2n+l-1\right)}^{2},\;(n=1,2,\;3\ldots ;\;l=0,\;1,\;2,\ldots )$$

It is important to remark that this equation becomes exact for $E(n,l)\to 0^{-}$(threshold conditions) [23]. Moreover, by assuming a liquid drop model for the clusters, so that $R=R(N)=2.22N^{1/3}$Ả, it is found that Equation (7) reproduces the threshold (vanishing eigenvalue) condition for (^3^He)*_N_* clusters (*N* = 40, 70, 112, 168) as calculated with a density functional [6] and the same value of $c\equiv b{\left(8m^{*}V_{0}\right)}^{1/2}/\pi \hbar =1.76\pm 0.03$. Since the evaporation probability, as shown below, depends essentially on the threshold quantum numbers, i.e., on the number of bound states, and much less on their detailed energy distribution, the following calculations based on the spherical-box model may be compared with the experimental data. For (^4^He)*_N_* clusters no similar check can be made with density-functional results. However, Equation (6) provides a good fit of the single-particle levels obtained from the variational Monte Carlo calculations by Lewart et al. for (^4^He)_70_ clusters [18] with *c* = 2.47 and a threshold number 2n+l−1 between 10 and 11. Moreover a convenient expression for *R*(*N*) can be fitted to Pandharipande et al. Green’s function Monte Carlo results [3]:(7)$$R(N)=b_{1}N^{1/3}+\frac{b_{2}}{N^{1/3}-1}-\frac{3\pi \hbar }{{\left(8m^{*}V_{0}\right)}^{1/2}}$$
with *b*_1_ = 1.88Å and *b*_2_ = 1.13Å. With *V*_0_ corresponding to the binding energy per atom for the bulk liquid (7.2 K [7]), an effective mass *m** = 3.2 × 4 a.u. is obtained. Here only values of *N* above the minimum of *R*(*N*) at about *N* = 7 shall be considered. It should be noted that *V*_0_ is actually a smooth function of *N* [7]. Assuming a constant *V*_0_, however, is affecting the amplitude of magic number maxima but essentially not their position, which is the scope of the present analysis. The effects of a *V*_0_(*N*) fitted to Guardiola et al. result [7] are discussed in Ref. [24].

The total Auger evaporation rate of one atom from an *N*-atom cluster is given by
(8)$$P(N)=\int dE\;\sum_{E_{1},E_{2},E_{2}\prime }\;W(E_{1},E_{2}\to E,\;E_{2}\prime )\;n(E_{1})n(E_{2})[1+n(E_{2}\prime )]$$
where $n(E)={\left[e^{(E-\mu_{B})/k_{B}T}-1\right]}^{-1}$ is the Bose-Einstein occupation number and the chemical potential $\mu_{B}\equiv \mu_{B}(T,N)$ is determined at each temperature from normalization to the atom number *N*. In the kinetic regime discussed above, controlled by single-atom evaporation, ρ(N)∝1/P(N).

It appears from various numerical tests that the integrated evaporation rate, Equation (8), is rather insensitive to the choice of *V*_0_ and *m** as long as the number of one-particle bound states remains the same, thus confirming that the threshold behavior, i.e., the insertion of new states, rather than the bound state spectrum, determines the stepwise dependence on *N* of the evaporation rate. In general terms this is consistent with the thermodynamic interpretation by Guardiola et al. [7] based on a diffusion Monte Carlo analysis of the *N*-dependent partition function, though their calculation refers to a larger temperature (1.7 K) and to collective excitations with smaller threshold numbers.

The calculations show that the distribution ρ(N)∝1/P(N) remains within the same order of magnitude for *N* varying from 10 to 100, whereas the size distribution function $\rho_{expt}(N)$ of the incident cluster beams, reported by Brühl et al. for different source pressures *P*_0_, decreases for increasing *N* over a few orders of magnitude [9]. The latter were obtained from the diffraction experiments via multiplication by a Jacobian factor and a factor correcting for the cluster ionization probability, altogether giving a factor $\propto N^{-3}$. To isolate and identify the magic-number oscillations, the actual $\rho_{expt}(N)$ is divided by its smoothed version, $\rho_{fit}(N)$, obtained from averaging over the oscillations, actually a power law, $\rho_{fit}(N)$ ∝ $N^{-\alpha }$ with *α* ranging from 1.74, for a cluster beam source pressure *P*_0_ = 1.33 bar, to 3.43 for *P*_0_ = 1.10 bar. The dependence of the exponent *α* on *P*_0_ clearly concerns the cluster formation kinetics close to the source, whereas the experiment indicates that the magic number oscillations are practically independent of *P*_0_ and are therefore linked with the kinetics inside the travelling beam.

In Figure 3 the experimental distributions $\rho_{expt}(N)/\rho_{fit}(N)$ as reported by Brühl et al. [9] are compared with the corresponding theoretical distributions for two different temperatures and *α* = 3. For a closer comparison at large *N* the calculated *ρ* (*N*) has been replaced by the Gaussian convolution
(9)$$\rho^{*}(N)=\frac{1}{N^{2}s\sqrt{\pi }}\sum_{N\prime }\rho (N\prime )\exp\left[-\frac{{\left(N\prime -N\right)}^{2}}{s^{2}N^{4}}\right]$$
in order to reproduce the finite instrumental angular resolution in the cluster number $\Delta N=sN^{2}$, where *s* ≅ 0.002.

The calculated distributions, plotted in Figure 2b for three different temperatures and *N* > 7, appear to reproduce very well all the salient features of the experiments. Some of the sharp peaks occurring in the calculated distributions are quite dependent on temperature: those at *N* = 11 and 21 are prominent at *T* = 0.37 K, but barely visible at 0.8 K, whereas the features for *N* > 40 increase with temperature. This can be interpreted as because clusters get smaller via more evaporation processes and are therefore favored at smaller temperatures. Experiments show a gradual switch of intensity from the peak at *N =* 21 to that at *N =* 26 for a decreasing source pressure *P*_0_, whereas theory shows a similar effect for increasing cluster temperature. 

The experimental oscillations in cluster abundance at larger *N* are better seen when plotted as a function of 1/*N* (Figure 3a). The calculated distribution for 0.8 K shows four peaks of comparable intensities: those at smaller N correspond to the experimental peaks for smaller source pressures, whereas those at larger N correspond to the experimental peaks at larger source pressures.

## 3. Conclusions

The good qualitative agreement between experiment and the calculated ^4^He cluster size distribution, although based on a simple free-particle model, provides a convincing argument in favor of the Auger evaporation mechanism for magic numbers (actually stability regions) of boson clusters. It is important to note that the Auger evaporation approach is conceptually similar to Guardiola et al. [6,7] quantum Monte Carlo thermo-dynamic approach in terms of threshold states, but they are alternative in terms of who is doing the job: single particle (actually pair collisions) or collective excitations?

Another issue concerning Brühl et al. experiments [9] is cluster formation kinetics. The observed similarity of peak distributions obtained at different source pressures after dividing out the respective smoothed distribution speaks in favor of in-beam kinetics. According to the present analysis Auger evaporation appears to be the basic kinetic mechanism for magic numbers in ^4^He clusters as generated in supersonic beams.

## Figures and Tables

**Figure 1 molecules-26-06244-f001:**
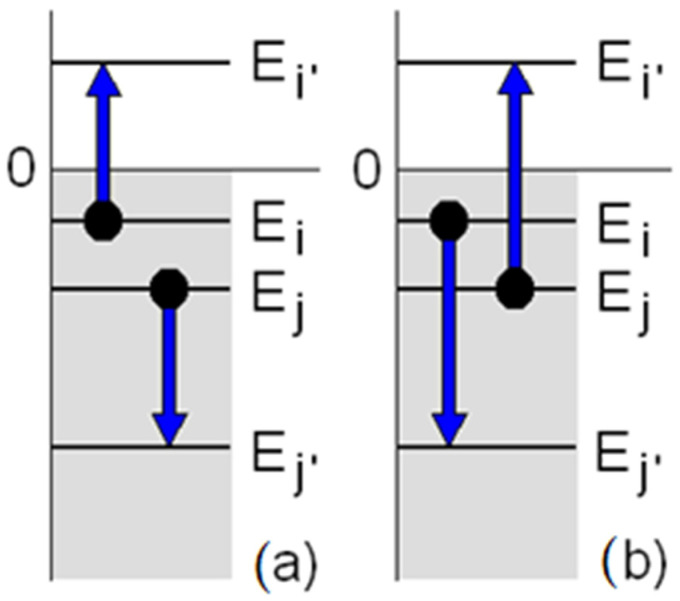
The Auger processes involving the collision of two atoms in the excited one-particle states *E_i_* and *E_j_* which lead to the emission into a free state *E_i’_* of either the atom in the state *E_i_* (**a**) or the atom in the state *E_j_* (**b**) with the other atom decaying into a lower state *E_j’_*.

**Figure 2 molecules-26-06244-f002:**
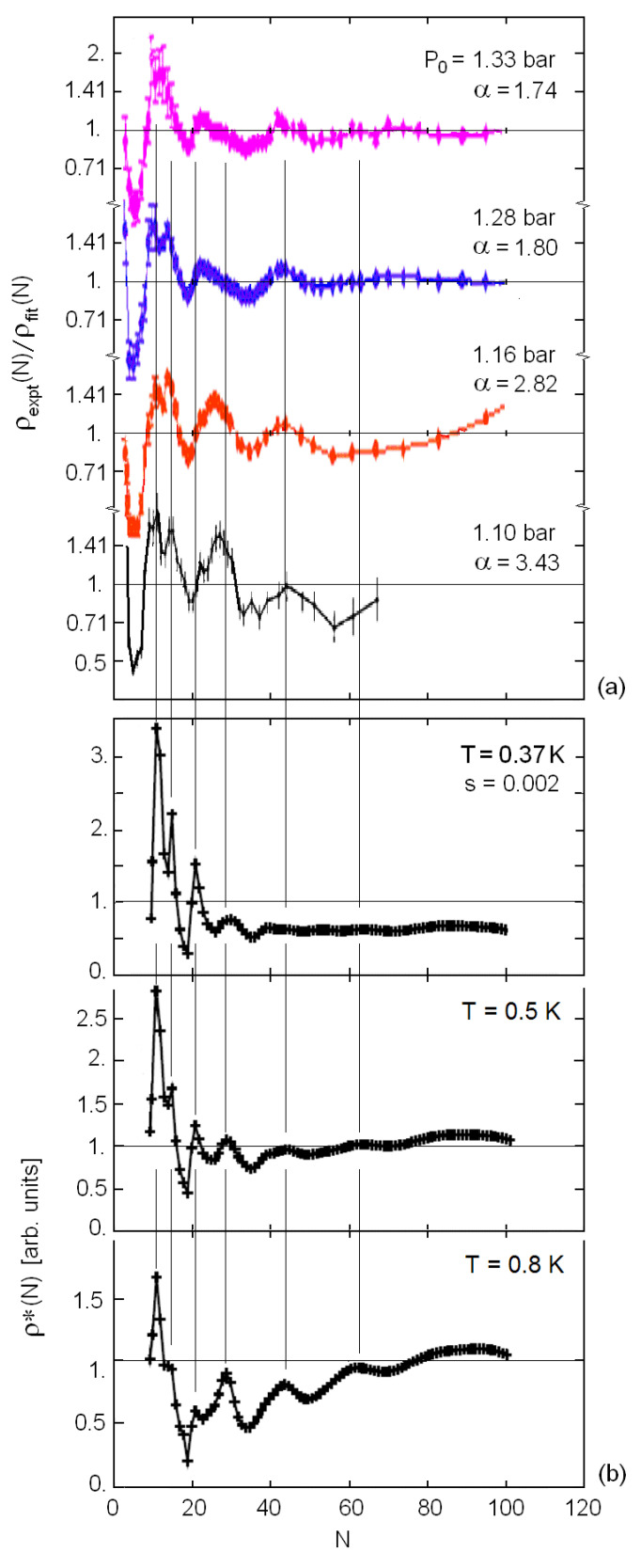
Comparison between (**a**) the (^4^He)*_N_* cluster size distributions obtained from cluster diffraction experiments by Brühl et al. [9] for different source pressures *P*_0_, and (**b**) the theoretical distributions for three different cluster temperatures and for an instrumental angular resolution parameter *s* = 0.002. Each experimental distribution is divided by a distribution $\rho_{fit}(N)\propto N^{-\alpha }$ obtained from averaging over the magic-number oscillations with *α* given by a best fit.

**Figure 3 molecules-26-06244-f003:**
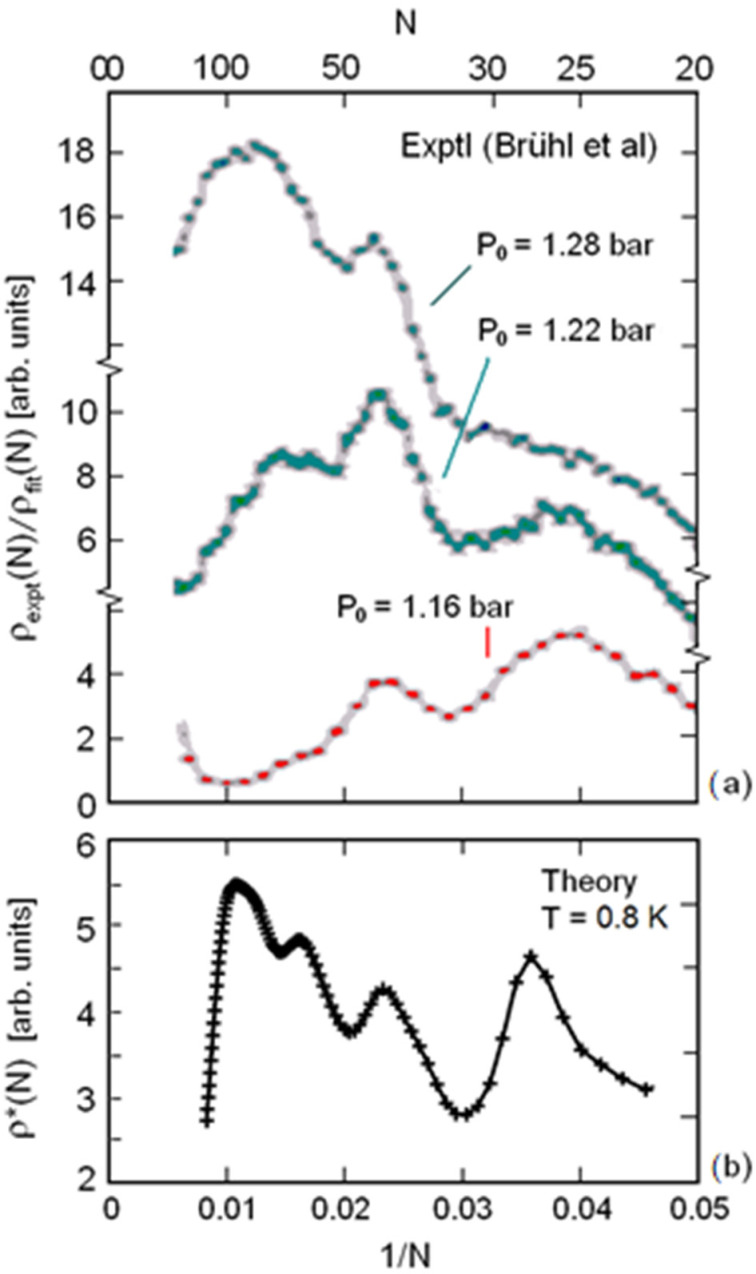
Same as Figure 2 with respect to a linear 1/*N* scale for a comparison between experiment and theory (*T* = 0.8 K only) at large values of *N*: (**a**) oscillations in cluster abundance and (**b**) the calculated distribution for 0.8 K.

## Data Availability

Detailed theoretical results reported in [24] may be requested to G.B.

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
