# Peer review of "Magic Numbers in Boson 4He Clusters: The Auger Evaporation Mechanism"

_molecules, 2021, doi:10.3390/molecules26206244_

Round 1

Reviewer 1 Report

The authors explain the occurrence of magic numbers in the size distribution of  bosonic 4He clusters using a simple independent particle model of the He clusters. They assume that the cluster size distribution results from two sequential processes. First is the growth of internally hot clusters followed by collisions between cluster atoms in excited states within the cluster leading to selective evaporation via an Auger process. An exchange-symmetric two-atom wave function is used to calculate the collision dynamics between boson pairs. The qualitative agreement between the calculated 4He cluster size distributions and the cluster size distributions from the diffraction experiments of Bruhl et al. provides a convincing argument in support of the Auger evaporation mechanism for the occurrence of magic numbers.

I found this work to be a very interesting approach to a difficult problem. Using an independent particle model to calculate the Auger collision rate is novel and deserves further investigation. 

I recommend its publication as is.

Author Response

We thank the referee for the careful reading and the nice and positive report. As suggested have fixed the English where needed and corrected a few typos.

Reviewer 2 Report

This is a reasonable contribution that provides a theoretical mechanism for "magic numbers" observed experimentally in clusters of He-4.  The model seems reasonable and agrees at least qualitatively (or arguably semi-quantitatively) with experiment.   I think that the scientific content is publishable in its present form. 

The one criticism that I had about the presentation is that it almost looks as if the equations have been typeset in MS Word and then pasted in as figures.  As a consequence, all of the math looks terrible and unprofessional.  The authors should typeset the manuscript in LaTeX using the template provided by the publisher.

Author Response

We thank the referee for the careful reading and the positive report. We have fixed the English as requested and corrected a few residual typos. As regards his remark

"The one criticism that I had about the presentation is that it almost looks as if the equations have been typeset in MS Word and then pasted in as figures.  As a consequence, all of the math looks terrible and unprofessional.  The authors should typeset the manuscript in LaTeX using the template provided by the publisher."

it seems the referee has received a bad copy of the manuscript. We are sorry for that. The copy returned to us by the editor looks very good and "professional". We have however slightly improved the readability of equations by inserting where needed a better spacing with respect to text and slightly reduced the size of in-text formulas to that of the text.